# Machine-Learning-Based Prediction of Land Prices in Seoul, South Korea

**Jungsun Kim [1], Jaewoong Won [2,3,*], Hyeongsoon Kim [4] and Joonghyeok Heo [5]**

[1] Real Estate Artificial Intelligence Research Institute, Seoul 06651, Korea; kimsun2018@gmail.com
[2] Department of Real Estate, Graduate School of Tourism, Kyung Hee University, Seoul 02447, Korea
[3] Department of Smart City Planning and Real Estate, Kyung Hee University, Seoul 02447, Korea
[4] Seoul Appraisal Co., Ltd., Seoul 06654, Korea; hanskim21@hanmail.net
[5] Department of Geosciences, University of Texas-Permian Basin, Odessa, TX 79762, USA; heo_j@utpb.edu
[*] Correspondence: jwon@khu.ac.kr; Tel.: +82-02-961-0544

**Abstract:** The accurate estimation of real estate value helps the development of real estate policies that can respond to the complexities and instability of the real estate market. Previously, statistical methods were used to estimate real estate value, but machine learning methods have gained popularity because their predictions are more accurate. In contrast to existing studies that use various machine learning methods to estimate the transactions or list prices of real estate properties without separating the building and land prices, this study estimates land price using a large amount of land-use information obtained from various land- and building-related datasets. The random forest and XGBoost methods were used to estimate 52,900 land prices in Seoul, South Korea, from January 2017 to December 2020. The models were also separately trained for different land uses and different time periods. Overall, the results revealed that XGBoost yields a higher prediction accuracy. Whereas the XGBoost models were more accurate on the 2020 data than on the 2017–2020 data when analyzing residential areas, the random forest models were more accurate on the 2017–2020 data than on the 2020 data. Further analysis will extend the prediction model to consider submarkets determined by price volatility and locality.

**Keywords:** land price; prediction modeling; machine learning; ensemble; random forest; XGBoost

## 1. Introduction

Real estate has few market participants because of its high value [1]. This characteristic of real estate causes an asymmetry in market information and reduces market efficiency [2]. The quick and accurate estimation of real estate values resolves this instability in the real estate market to a certain degree and helps establish real estate policies [3]. For these reasons, attempts have been made to increase the quality of data in the public and private sectors to improve the estimation of real estate values, increase the efficiency and accuracy of valuation, and build an automated valuation model [4].

Recently, large-scale data collection and machine/deep-learning-based valuation models have become available in the real estate field owing to the application of information and artificial intelligence technologies. A machine learning approach can consider many variables and employs flexible, data-driven model specifications [5]. Previous empirical studies have verified these advantages in prediction performance when compared with the performance of traditional hedonic regression approaches. Simlai (2021) estimated the owner-occupied housing values of 7904 census tracks in California by employing least-squares-based machine-learning models such as ridge regression, least absolute shrinkage and selection operator (LASSO), and elastic net regression [6]. The study also utilized conventional ordinary least-squares (OLS) and weighted least-squares regression models. However, the results revealed that the machine learning models (ridge, LASSO, and elastic net) had better prediction capabilities than conventional regression models.

Schulz and Wersing (2021) used the data of 11,908 housing transactions from 2016 to 2019 in Aberdeen, Scotland, to evaluate boosting-based machine learning models, which yielded better prediction accuracy than conventional statistical models such as polynomial and spatial autoregressive models [7].

Previous studies have also compared the prediction capabilities of machine learning models. Mullainathan and Spiess (2017) estimated 10,000 owner-occupied housing prices randomly selected from the 2011 American Housing Survey by employing a tree-based ensemble machine learning as well as regression-based machine learning [8]. They incorporated 150 various covariates to perform the estimation and concluded that the prediction capability of the random forest (RF), which is a typical ensemble method, was superior. Ceh et al. (2018) estimated 7407 apartment sale prices in the city of Ljubljana, Slovenia using RF and OLS methods and showed better performances in terms of R-squared using the RF method [9]. Singh et al. (2020) used housing sales data from 2006–2010 in Ames, Iowa, USA, to estimate prices using the RF, gradient boosting, and LASSO machine learning techniques [10]. Their study showed that the prediction accuracy was high when prices were estimated by gradient boosting. Pai and Wang (2020) evaluated machine learning models (least-squares support vector regression, classification and regression trees, and backpropagation neural networks) with respect to the prediction of 32,215 housing prices from April 2016 to April 2019 in Taichung, Taiwan [11]. Their study considered 23 housing and environmental features and showed that the least-squares support vector regression model outperformed the others. Park and Bae (2015) analyzed classification-based machine learning models such as the C4.5 DT algorithm, repeated incremental pruning to produce error reduction (RIPPER), naïve Bayes method, and AdaBoost using the list price data of 5359 townhouses from the Multiple Listing Service in Fairfax County, VA, USA [12]. Their study considered 27 features such as structural characteristics (e.g., bathroom, bedroom, exterior features, heating, lot size, and parking), financial characteristics (e.g., mortgage), and public school ratings. The numerical results of their study showed that the RIPPER prediction model was the most accurate. Antipov and Pokryshevskaya (2012) used the price data of 2848 two-room apartment transactions completed in 2010 in Saint-Petersburg, Russia, to estimate prices using various machine learning methods such as RF, k-nearest neighbors, boosting, the classification and regression tree, chi-squared automatic interaction detection, and an artificial neural network [13]. Alfaro-Navarro et al. (2020) used different ensemble methods (bagging, RF, and boosting) to estimate 790,631 real estate prices with 33 feature variables representing property characteristics in Spain [14]. Their results showed better performances in terms of the mean absolute percentage error in the bagging and RF methods. Ho et al. (2021) used 39,554 housing prices from 1996 to 2014 in Hong Kong to analyze machine-learning models for price estimation [15]. Their study used support vector machine, RF, and gradient boosting machine (GBM) models to estimate the prices, and the analysis results verified that the accuracies of RF and GBM were higher than that of the support vector machine. A study by Truong et al. (2020) used RF, extreme gradient boosting (XGBoost), light gradient boosting machine, hybrid regression, and stacked generalization regression models to estimate more than 300,000 housing prices in Beijing, China, with 58 features and found that the RF model achieved the lowest root mean squared logarithmic error [16].

Although the performance of machine learning methods in previous studies varied according to the study domain and dataset, the overall high accuracy of machine learning models (both bagging-based (e.g., RF) or boosting-based (e.g., GBM) ensemble approaches) has been generally verified. Previous studies have broadened our understanding of the application of machine learning techniques to real estate value estimation, but the following three research gaps remain.

First, most previous studies estimated the transaction or list prices of real estate without separating land and structural components. The final real estate value is based on the combination of the contributions of land and improvement components, but these two contributions are not separately listed in normal market transactions. However, the

estimated value of the land alone might provide information that is important for real estate businesses and policies. By estimating the value of the land only, the feasibility of an investment can be more efficiently determined. In addition, the valuation of the land provides an important basis for establishing real estate taxation policies, as in the example of South Korea. Buildings deteriorate over time, which causes a depreciation in their prices and leads to errors in real estate valuations. Davis and Heathcote (2007) noted that improvement values are estimated as the depreciation of construction costs [17]. Previous research also found that land price leads to more volatile trends in the real estate market compared with housing price [17,18], and different factors determine land and improvement components [19]. The extensive hedonic price literature has shown housing prices are a function of various factors ranging from structural (e.g., built year, lot size, and number of rooms) and financial characteristics (e.g., foreclosed status) to neighborhood (e.g., socioeconomic, safety, built environments) characteristics [20,21]. Land price has long relied on monocentric models mainly explained by accessibility (e.g., distance to the central business district) and land use density as essential concepts in land economic theories [22,23]. Previous studies found significant impacts of accessibility to jobs and nearby amenities such as park, open space, and waters [24–26], but studies rarely explored land use factors, which are more likely to influence land price in modern developed urban environments.

Second, a system for collecting land price data and modeling automated valuations needs to be built. Zillow, a representative American real estate platform (zillow.com), collects various real estate related information on topics such as population, school districts, crimes, geography, multiple listing services, and sale and lease transactions. It also provides predicted prices obtained from machine-learning techniques under the name of Zestimate. In South Korea, several platforms that provide real estate price information, such as Zigbang (zigbang.com), Dabang (dabangapp.com), and Value Map (valueupmap.com) are available; nonetheless, these platforms provide information only on residential real estate prices or brokerage services. Recently, the government has provided massive information related to real estate in the form of an open data platform via an application programming interface (API) to encourage its use. Although data collection through an API can overcome the limitations of conventional data collection methods, such as manual collection or collection that requires the permission of real-estate agents, a process for integrating and refining various real estate data is still needed to extract adequate information about land transaction prices. Hence, in this study, the process of data collection was automated using a patented technique developed by Seoul Appraisal Co., Ltd., and rich datasets to analyze land prices.

Third, an insufficient number of studies have been conducted on real estate values in Seoul. Seoul is the capital city of South Korea, has a population of 9.8 million people, and a population density of 26,000 people/km$^2$. It is large enough to provide the number of real estate transaction samples required for analyses according to various socioeconomic activities. Recent studies analyzed machine learning models that estimated real estate values in Seoul; however, their study was limited to the valuation of apartments [4] or buildings [27]. Accordingly, this study constructed a dataset consisting of the land transaction prices of 52,900 lots in Seoul from 2017 to 2020 and employed ensemble-based RF and eXtreme gradient boosting (XGBoost), both high-performance machine learning methods, to estimate land prices using data that contain rich information on land use. The results of this study demonstrate the potential for a more sophisticated prediction model.

The rest of this paper is structured as follows. The next section elaborates on the study methodology, including the descriptions of the study area, the data sources, the variable measures, and two modeling techniques, Random Forest and XGBoost. Thereafter, the summary statistics and the empirical results are presented and compared in Section 3. The interpretation of the two empirical models and their associations with various land use variables are discussed in Section 4. Conclusions are made in the closing section.

## 2. Materials and Methods

### 2.1. Study Area

The study area is Seoul, South Korea, which has an area of 605 km$^2$ and a population of 9.8 million people. Seoul consists of 25 boroughs (called "gu"), 425 administrative districts (called "dong"), and 640,575 parcels of property. Each borough has 25,623 parcels on average, ranging from 13,109 parcels to 39,010 parcels. Seoul includes land for various uses (e.g., residential, commercial, industrial, and green) and natural environments (e.g., rivers and mountains). Commercial, residential, and green land uses account for 5.9%, 18.9%, and 41.8% of the total area of Seoul, respectively. A mixture of residential and commercial land use accounts for 13% of the total area. The Han River runs through the city, and green belts exist in the outskirts of the city. The study area is shown in Figure 1.

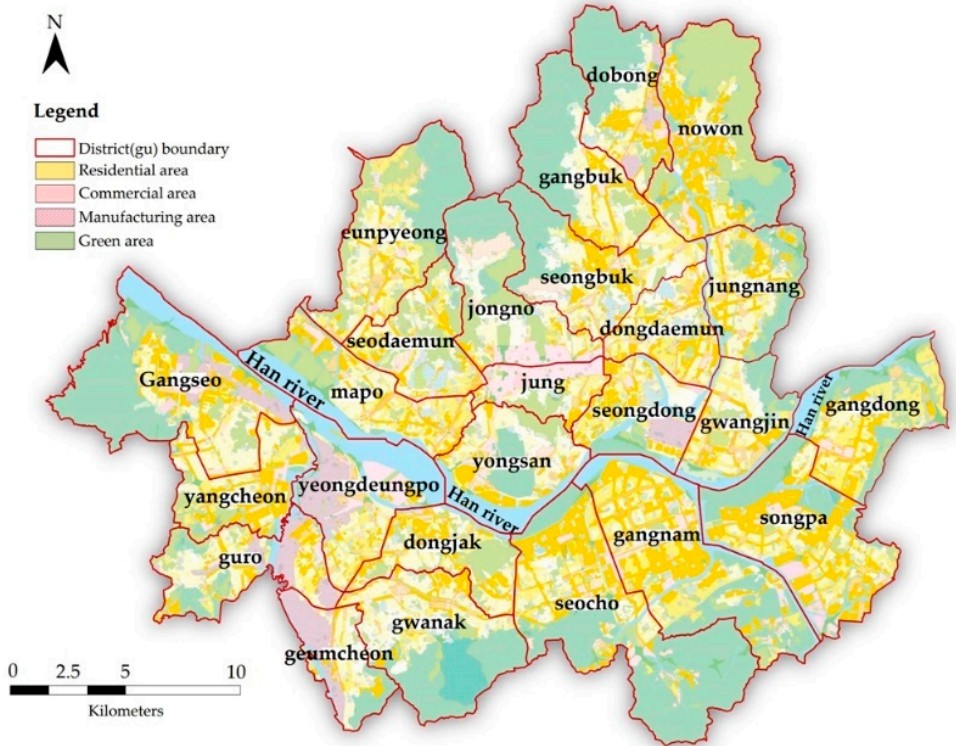

**Figure 1.** Study area.

### 2.2. Data Sources

The data were obtained from government websites operated by the Ministry of Land, Infrastructure, and Transport (MLIT) and the Korea Real Estate Board (KREB). The MLIT is a central administrative agency responsible for establishing policies and laws related to the national territory, construction, and real estate. The KREB is a public enterprise of the MLIT that discloses real estate prices and performs the analysis and management of real estate information.

The public APIs provided by the MLIT were used to collect information from the building register, land use, appraised land value, real estate transaction price, and land price change rate datasets. The dataset of the standard unit prices of buildings was purchased from the KREB. The details of the datasets are described in Section 2.3. All datasets except for the real estate transaction price dataset were merged according to parcel number to form a single dataset for the analysis. Because the datasets are provided to the public, the real estate transaction price data do not include the parcel numbers or specific addresses because of privacy concerns. To merge the transaction price data, we used a matching algorithm patented by Seoul Appraisal Co., Ltd. (Patent No. 10-1857011, Korean

Intellectual Property Office). The steps used to process datasets and variables are illustrated in Figure 2.

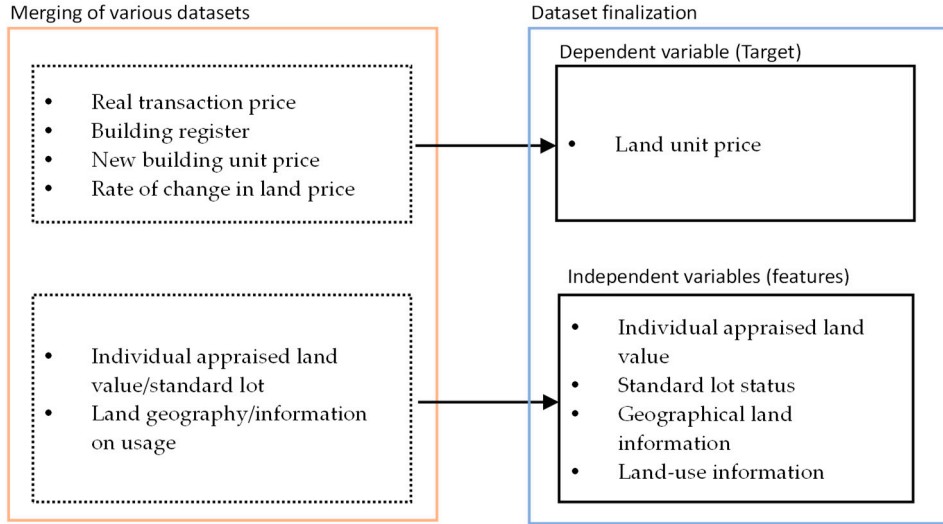

**Figure 2.** Dataset and variable processing steps.

In this study, all transaction data generated from January 2017 to December 2020 were collected. A total of 60,456 arm's length transactions occurred during this period. The transactions for apartments and small sites (less than 3.3 m$^2$) were excluded. The data of 52,900 transactions were used for our final analysis. Table 1 shows the number of transactions by land use in Seoul for each year. The residential areas of Seoul make up approximately 90% of all transactions for each year, and the commercial areas, industrial areas, and green areas make up the rest, in that order.

**Table 1.** Number of transactions by land use from 2017 to 2020 in Seoul (unit: m$^2$).

|  | Residential Areas | Commercial Areas | Industrial Areas | Green Areas | Total |
|---|---|---|---|---|---|
| 2017 | 14,078 | 662 | 328 | 236 | 15,304 |
| 2018 | 12,448 | 784 | 312 | 256 | 13,800 |
| 2019 | 10,044 | 825 | 304 | 213 | 11,386 |
| 2020 | 11,085 | 823 | 283 | 219 | 12,410 |
| Total | 47,655 | 3094 | 1227 | 924 | 52,900 |

*2.3. Variables*

The independent variables (or features) were based on the land use and appraised land value data. The land use data included both geographical and land use information. Various geographical attributes corresponding to administrative location (e.g., borough and district), bearing (i.e., east, west, south, north, southeast, northeast, or northwest), area size, topography (e.g., steep slopes), shape (e.g., irregular or square), and road interface (e.g., distributor or arterial roads) were measured. Various land use attributes corresponding to the main zoning (e.g., residential or commercial), special district designation, land category, planned facilities, area restrictions, farmland classification, forest land, railroads, and waste were also measured. The appraised land value data were used to measure the appraisal land value and determine the standard lot status.

The dependent variable (or target) of this study was the land unit price. Land unit prices were calculated as the land transaction price divided by land area. This study used four-year transaction price data from 2017 to 2020, and the rate of change in land value from the transaction date to 31 December 2020 was applied to the land unit price at transaction time to obtain a value that is equivalent to the price as of 31 December 2020. For example, the land unit price as of 31 December 2020 was calculated by multiplying the land unit price at the transaction date by the rate of change in land value from the corresponding transaction date to 31 December 2020. The calculation formulas are as follows.

1.  Building price = Replacement cost − Depreciation amount (applied from the approved date of use to the transaction time);
2.  Land unit price at transaction time = (real transaction price of the real estate − Building price)/land area;
3.  Land unit price as of 31 December 2020 = land unit price at transaction time × rate of change in land price (from the transaction time to 31 December 2020).

The real estate transaction price data from the MLIT provided real estate prices without separate prices for the land and buildings. If a building exists on a parcel, the building price was subtracted from the transaction price to obtain the land price. The building price was calculated by subtracting the depreciation amount from the cost of replacement (or reproduction). The amount of depreciation was calculated based on a valuation of the wear or loss over time from the approved date of use to 31 December 2020. The replacement cost refers to the reasonable cost required to reproduce or reacquire a target building in new-construction condition for full utilization at the current market price. The data for the standard unit price of new buildings included the criterion for calculating the replacement cost, which is determined according to the building structure and its use. The building register data included the details about the building structure (e.g., wood, block, or masonry) and building use (e.g., residential, commercial, or industrial).

### 2.4. Analysis

2.4.1. Analytical Framework

We conducted a three-fold analysis. The real estate market often varies according to temporal and spatial submarket characteristics. We first separated our final dataset into 2017–2020 data and 2020 data to examine how the prediction of land prices changed according to temporal changes in the market. Second, we separated the dataset based on land use to examine how the prediction of land prices changed according to land-use differences. Third, we examined the prediction of land prices with and without the appraised land value variable. All analyses were carried out using Python 3.7 and the relevant program packages.

2.4.2. Machine Learning Methods: RF and XGBoost

This study employed the ensemble approach methods of RF and XGBoost, which have been found to yield high performance in both previous studies and Kaggle competitions. The ensemble approach combines weak multiple predictors to create a single strong predictive model. This single ensemble model aggregates the results derived from all predictors and often yields a higher accuracy than any individual predictor model [28]. The predictors are broadly trained in three ways to predict the class: voting, bagging, or boosting. In the voting method, the predictors are independently trained, and the final predicted target class is determined via majority vote. The bagging method also employs majority vote predictors, but sampling with replacement is used for each predictor, which has been trained using the same training algorithm. In the boosting method, multiple predictors are sequentially trained, and each predictive model is corrected using information from the predecessor. RF is trained using the bagging method, which combines multiple independent decision tree (DT) based classifiers [29]. XGBoost is trained using the boosting method and consists of sequential classifiers using DTs as the base predictors to correct the errors of the preceding, underfitted predictor. A DT is a tree-like structure in which

a root node, which forms the initial node of the tree structure, is split into internal nodes representing a decision on feature classification, and a final class label is assigned to a leaf node, which represents the final target value [30]. The classification rules for achieving high data uniformity are applied to the data until the nodes cannot be further divided into sub-nodes [30]. The uniformity of the final classified data is measured using the Gini or entropy index to select the best DT model [30].

The RF model is built using randomly selected input variables and sampling with replacement [31]. We followed a three-step procedure for using the RF model: (1) the DTs are generated using bootstrap sampling; (2) the hyperparameters, such as the number of trees, depth of the trees, and number of randomly selected input variables, are tuned; (3) the RF model is trained and used to predict the final values by voting. The hyperparameters were determined using the GridSearch function in the scikit-learn Python library. GridSearch generates a grid of hyperparameter values and specifies which hyperparameters to use to control the learning process [32]. The number of input variables was set to five, following previous empirical studies [33]. The RandomForestRegressor algorithm in scikit-learn was used to analyze the RF model. The hyperparameters were set as follows: 10,000 for the number of trees (n_estimators), 9 for the depth of trees (max_depth), and default values for all other hyperparameters. The details on the hyperparameters are presented in Table 2.

**Table 2.** Hyperparameters of the algorithms used in the study.

| Model | Python Library | Hyperparameter |
|---|---|---|
| Random Forest | RandomforestRegressor from Scikit-Learn | n_estimators = 10,000, max_depth = 9, and default for others (n_estimators = 100, criterion = 'mse', max_depth = None, min_samples_split = 2, min_samples_leaf = 1, min_weight_fraction_leaf = 0.0, max_features = 'auto', max_leaf_nodes = None, min_impurity_decrease = 0.0, min_impurity_split = None, bootstrap = True, oob_score = False, n_jobs = None, random_state = None, verbose = 0, warm_start = False, ccp_alpha = 0.0, max_samples = None) |
| XGBoost | XGBoostRegressor from Scikit-Learn Wrapper | n_estimators = 18,385, max_depth = 6, learning_rate = 0.005, obj = squarederror, and default for others (base_score = 0.5, booster = gb_tree, colsample_bylevel = 1, colsample_bynode = 1, colsample_bytree = 1, gamma = 0, importance_type = 'gain', max_delta_step = 0, min_child_weight = 1, missing = None, nthread = $-1$, reg_alpha = 0, reg_lambda = 1, scale_post_weight = 1, seed = 0, subsample = 1, verbosity = 1) |

XGBoost is a scalable model that reduces overfitting and bias and increases computational speed and model performance by applying gradient boosting-based tree pruning, parallelization in tree construction, and regularization [34]. The fast runtime of XGBoost is achieved by applying cache-aware access and out-of-core computation [34]. The boosting approach creates a strong learner by combining multiple weak learners with a simple tree structure. When a weak learner is trained at each iteration, the weights are updated to the next learner to increase the prediction performance and reduce the bias. This method employs the gradient descent algorithm to reduce residuals resulting from the predecessor learner. The XGBoost algorithm in the scikit-learn wrapper module was used [35]. The hyperparameters were determined to be n_estimators = 18,385, max_depth = 6, learning_rate = 0.005, and default values were used for the remaining hyperparameters (see Table 2).

### 2.4.3. Model Evaluation Measure

The data were randomly split into training, validation, and test sets using a 7:1:2 ratio, which is a common technique used to evaluate a model [32]. For model evaluation, we used the accuracy rate, which is calculated as the number of correct predictions. Prediction was regarded as being correct when the predicted land price was within the ±10% error

margin of the actual land price. We used this range because it has been widely accepted for valuation errors [36,37]. We also tested 5% and 15% error margins, but for brevity, only the results with 10% error margin are reported in this paper.

- Prediction is correct if $\left| \frac{\text{predicted land price} - \text{actual land price}}{\text{actual land price}} \right| \leq 10\%$

- Accuracy $= \frac{\text{Number of correct predictions}}{\text{Total number of predictions}}$

## 3. Results

### 3.1. Summary Statistics

Table 3 summarizes the statistics of the variables. The average land unit price of the samples was KRW 8,109,860. The average appraised land value was KRW 4,466,413. The samples had an average area of 199.931 m$^2$. They were mostly flatland or elevated areas, had the shape of a square, rectangle, or ladder, and abutted onto medium-sized or narrow roads. The main zoning was residential, and most areas were not designated as restricted areas or specific use areas.

**Table 3.** Summary of statistics.

| Variables | Descriptions | Mean/ Frequency | S.D. (Min.–Max.)/ % |
|---|---|---|---|
| *Dependent variable (Target)* | | | |
| Land unit price | Continuous: (KRW) | 8,109,860 | 7,322,789 (9201–326,671,182) |
| *Independent variables (Features)* | | | |
| *Appraisal Information* | | | |
| Appraised land value | Continuous: (KRW) | 4,466,413 | 3,785,368 (7240–176,000,000) |
| Standard lot status | Binary: 1: standard lot 0: non-standard lot | 2403 50,497 | 4.543% 95.457% |
| *Geographical Land Information* | | | |
| Area | Continuous: m$^2$ | 199.931 | 992.714 (3.3–177435) |
| Topography | Category: 1: Steep slope 2: Undulating slope 3: Flatland 4: Low-lying area 5: Elevated area | 393 1275 10,749 38 40,445 | 0.743% 2.410% 20.319% 0.072% 76.456% |
| Shape | Category: 1: Irregular 2: Square 3: Ladder 4: Triangle 5: Flag 6: Vertical rectangle 7: Horizontal rectangle 8: Inverted triangle | 4447 8577 16,059 828 3199 13,620 6147 23 | 8.406% 16.214% 30.357% 1.565% 6.047% 25.747% 11.620% 0.043% |
| Abutting road | Category: 1: Thoroughfare 2: Medium-sized road 3: Medium-narrow road 4: Narrow road 5: Land with no road access | 3142 3096 6971 39,267 424 | 5.940% 5.853% 13.178% 74.229% 0.802% |
| *Land Use Information* | | | |
| First main zoning | Category: 1: Residential 2: Commercial 3: Industrial 4: Green | 47,655 3094 1227 924 | 90.085% 5.849% 2.319% 1.747% |
| Area of first main zoning | Category: 1: Residential 2: Commercial 3: Industrial 4: Green | 163.830 231.112 241.188 1687.738 | 211.528 (3.4–14,149) 1037.090 (3.4–49,206) 543.344 (4–8209) 6912.444 (5–177,435) |
| Second main zoning | Category: 1: Residential 2: Commercial 3: Green | 793 61 55 | 1.499% 0.115% 0.104% |
| Area of second main zoning | Category: 1: Residential 2: Commercial 3: Industrial 4: Green | 0.7658 13.710 0.726 38.113 | 15.649 (0–1920) 91.575 (0–2630) 15.650 (0–478) 869.074 (0–2598) |

**Table 3.** *Cont.*

| Variables | Descriptions | Mean/<br>Frequency | S.D. (Min.–Max.)/<br>% |
|---|---|---|---|
| Restricted area | Binary: 1: Restricted area<br>0: Non-restricted area | 1372<br>51,528 | 2.594%<br>97.406% |
| Specific use area | Binary: 1: Specific use area<br>0: Non-specific use area | 9623<br>43,277 | 18.191%<br>81.809% |
| Forest land | Binary: 1: Forest land<br>0: other than forest land | 269<br>52,631 | 0.509%<br>99.491% |
| Farmland | Binary: 1: farmland<br>0: other than farmland | 274<br>52,626 | 0.518%<br>99.482% |
| Waste | Binary: 1: waste<br>0: other than waste | 40,846<br>12,054 | 77.214%<br>22.786% |
| Planned facilities | Binary: 1: planned facilities<br>0: other than planned facilities | 3898<br>49,002 | 7.369%<br>92.631% |
| Planned facility conflict rate | Continuous: % | 47.44 | 53.22 (0–100) |
| Land category | Category: 1: Park site<br>2: Orchard<br>3: Rice paddy<br>4: Site<br>5: Forest land<br>6: Miscellaneous land<br>7: Factory site<br>8: Field (dry)<br>9: Site for religious use<br>10: Gas station land<br>11: Parking site<br>12: Storage site<br>13: Right of way<br>14: Site for athletics use<br>15: School site | 4<br>4<br>274<br>51,518<br>348<br>90<br>65<br>451<br>30<br>29<br>30<br>3<br>31<br>1<br>22 | 0.008<br>0.008<br>0.518<br>97.388<br>0.658<br>0.170<br>0.123<br>0.853<br>0.057<br>0.055<br>0.057<br>0.006<br>0.059<br>0.002<br>0.042 |
| Distance to railway land | Category: 1. Within 10 m<br>2. Within 50 m<br>3. Within 100 m<br>4. Within 500 m<br>5. Beyond 500 m | 2443<br>4380<br>9953<br>17,861<br>18,263 | 4.62<br>8.28<br>18.81<br>33.76<br>34.52 |
| Land use details | Category: 1: Industrial<br>2: Orchard<br>3: Residential<br>4: Commercial<br>5: Farmland<br>6: Residential and commercial<br>complex<br>7: Office<br>8: Forest land | 256<br>5<br>37,004<br>7519<br>542<br>6781<br>556<br>237 | 0.484%<br>0.009%<br>69.951%<br>14.214%<br>1.025%<br>12.819%<br>1.051%<br>0.448% |

Note: Year dummy variables (2017, 2018, 2019, and 2020), administrative location dummy variables (borough and district), and geographic location dummy variables (east, west, south, and north) were included but not reported for brevity.

### 3.2. Empirical Results: Prediction Modeling

Table 4 provides the empirical results obtained by the RF and XGBoost models for predicting land unit prices in Seoul. The accuracy rates for models that consider different samples (the 2017–2020 sample and the 2020 sample) and variables are reported. The first model (M1) considers only the appraisal value variable, the second model (M2) considers all variables except for the appraisal land value variable, and the third model (M3) considers all variables. Overall, the XGBoost models yielded higher accuracy rates than the RF models. For RF models M1 and M2, the accuracy on the 2017–2020 sample was higher than it was on the 2020 sample, whereas the accuracy rates obtained by all XGBoost models on the 2020 sample were higher than those on the 2017–2020 sample. A larger sample might be expected to have better prediction capabilities, but differing results were obtained by the RF and XGBoost models. Model M3, which includes all variables, obtained higher accuracy rates than models M1 and M2.

**Table 4.** Empirical results of RF and XGBoost for predicting the land unit prices in Seoul.

| Year | Data | RF | | | XGBoost | | |
|---|---|---|---|---|---|---|---|
| | | **M1** | **M2** | **M3** | **M1** | **M2** | **M3** |
| 2017–2020 (N = 52,900) | Training | 76.46 | 45.30 | 78.39 | 85.91 | 55.09 | 88.96 |
| | Test | 75.42 | 45.30 | 77.41 | 84.03 | 56.19 | 87.82 |
| | All | 75.98 | 45.98 | 77.93 | 83.46 | 53.99 | 86.50 |
| 2020 (N = 12,410) | Training | 76.29 | 53.49 | 73.87 | 88.00 | 56.30 | 90.95 |
| | Test | 73.81 | 51.78 | 76.67 | 84.51 | 55.98 | 90.44 |
| | All | 74.98 | 51.99 | 74.28 | 85.86 | 54.58 | 89.76 |

Note: RF: random forest; M1 = model using only the appraised land value; M2 = model with all features except for the appraised land value; M3 = model with all features.

Table 5 presents the results of the prediction modeling by zoning. A comparison of models M1, M2, and M3 reveals that, as for the results of the entire sample, the accuracy of M3, which considers all the features, is higher overall for each zoning type. When analyzing the 2017–2020 sample, a comparison of the results of the models that incorporate all features showed that the accuracy rates of the RF model were high in the residential areas (78.55), industrial areas (76.84), commercial areas (73.93), and green areas (62.69) in that order, whereas the accuracy rates of XGBoost were high in the industrial areas (86.90), green areas (84.99), commercial areas (83.91), and residential areas (79.29) in that order. When analyzing only the 2020 data, it was found that the accuracy rates of the RF model were high in the residential areas (78.28), commercial areas (77.49), industrial areas (74.91), and green areas (62.87) in that order, whereas the accuracy rates of XGBoost were high in the industrial areas (85.88), residential and commercial areas (84.99), and green areas (82.58) in that order.

**Table 5.** Results of the prediction modeling by zoning.

| Year | Zones | Data | RF | | | XGBoost | | |
|---|---|---|---|---|---|---|---|---|
| | | | **M1** | **M2** | **M3** | **M1** | **M2** | **M3** |
| 2017–2020 | Residential (N = 47,655) | Training | 78.48 | 47.22 | 80.49 | 78.58 | 49.28 | 80.89 |
| | | Test | 76.57 | 45.66 | 78.04 | 75.92 | 45.74 | 78.70 |
| | | All | 77.44 | 46.30 | 78.55 | 76.19 | 47.94 | 79.29 |
| | Commercial (N = 3094) | Training | 70.33 | 58.29 | 72.98 | 83.99 | 68.98 | 84.09 |
| | | Test | 68.93 | 57.57 | 73.44 | 82.69 | 66.06 | 83.78 |
| | | All | 67.89 | 58.50 | 73.93 | 83.00 | 66.91 | 83.91 |
| | Industrial (N = 1227) | Training | 77.84 | 62.99 | 76.83 | 86.04 | 73.91 | 87.95 |
| | | Test | 75.80 | 60.70 | 77.56 | 84.98 | 71.09 | 85.38 |
| | | All | 76.84 | 61.98 | 76.84 | 85.98 | 72.98 | 86.90 |
| | Green (N = 924) | Training | 59.30 | 57.30 | 61.98 | 83.98 | 75.99 | 85.09 |
| | | Test | 58.00 | 55.97 | 63.33 | 81.56 | 73.45 | 83.58 |
| | | All | 58.30 | 56.49 | 62.69 | 82.99 | 74.99 | 84.99 |
| 2020 | Residential (N = 11,085) | Training | 77.49 | 55.48 | 79.39 | 85.91 | 55.98 | 86.09 |
| | | Test | 74.51 | 52.65 | 77.27 | 83.37 | 52.83 | 84.85 |
| | | All | 76.49 | 53.30 | 78.28 | 84.12 | 53.99 | 84.99 |
| | Commercial (N = 823) | Training | 72.33 | 66.49 | 78.30 | 83.99 | 77.86 | 85.90 |
| | | Test | 70.48 | 62.81 | 76.27 | 82.76 | 76.03 | 83.83 |
| | | All | 71.98 | 64.30 | 77.49 | 82.99 | 76.55 | 84.99 |
| | Industrial (N = 283) | Training | 74.56 | 66.24 | 75.99 | 85.99 | 78.79 | 86.99 |
| | | Test | 73.70 | 64.94 | 73.38 | 83.77 | 77.60 | 85.71 |
| | | All | 74.29 | 65.49 | 74.91 | 84.98 | 78.18 | 85.88 |
| | Green (N = 219) | Training | 56.13 | 55.53 | 64.92 | 80.20 | 79.49 | 81.98 |
| | | Test | 54.51 | 53.65 | 61.37 | 79.83 | 77.25 | 80.69 |
| | | All | 55.39 | 54.39 | 62.87 | 79.50 | 78.72 | 82.58 |

Note: RF: random forest; M1 = model using only the appraised land value; M2 = model with all features except for the appraised land value; M3 = model with all features.

Table 6 shows the results from an assessment of the importance of the variables that determine land unit prices. The appraisal land value was the most influential factor in both the RF and XGBoost models, but the importance of other variables varied; nevertheless, it was found that when considering the top 10 variables ordered by importance, the geographical features of the area and administrative location, as well as the land-use features of land use, main zoning, and special district, appeared to be important in predicting the land prices.

**Table 6.** Variables that determine land unit prices ranked by importance.

| Ranking | RF | XGBoost |
|:---:|:---:|:---:|
| 1 | Land appraisal value (0.160) | Land appraisal value (0.240) |
| 2 | Area (0.112) | Land use (0.153) |
| 3 | Main zoning area (0.107) | Dong (0.101) |
| 4 | Road condition (0.081) | Main zoning (0.087) |
| 5 | Land use (0.073) | Gu (0.049) |
| 6 | Dong (0.071) | Specific use district (0.041) |
| 7 | Main zoning (0.058) | Second zoning area (0.041) |
| 8 | Shape (0.058) | Second zoning (0.033) |
| 9 | Land category (0.058) | Road condition (0.031) |
| 10 | Bearing (0.039) | Accessibility to waste facilities (0.022) |
| 11 | Restrictions (0.034) | Restrictions (0.021) |
| 12 | Area ratio included (0.030) | Bearing (0.020) |
| 13 | Urban planning facilities (0.025) | Reference lot (0.017) |
| 14 | Accessibility to waste facilities (0.020) | Land category (0.016) |
| 15 | Agricultural land (0.020) | Area (0.015) |
| 16 | Distance to railway land (0.014) | Urban planning facilities (0.014) |
| 17 | Topography (0.013) | Main zoning area (0.014) |
| 18 | Reference lot (0.010) | Distance to railway land (0.014) |
| 19 | Specific-use district aea (0.006) | Year (0.013) |
| 20 | Forest land (0.004) | Area ratio included (0.013) |
| 21 | Second zoning (0.004) | Agricultural land (0.012) |
| 22 | Second zoning area (0.003) | Topography (0.011) |
| 23 | Gu (0.001) | Shape (0.011) |
| 24 | Year (0.001) | Forest land (0.010) |

## 4. Discussion

In this study, a practical evaluation of the use of machine learning techniques to predict land prices was performed, which has rarely been considered in previous studies. We calculated land unit prices and diverse land use variables from various datasets obtained via public APIs provided by government websites. The data were predicted using ensemble-based RF and XGBoost models, which have been found to be machine learning methods with excellent prediction capabilities.

The results of this study showed that the prediction accuracies of the XGBoost models were overall higher than those of the RF models. In the reviewed literature, housing prices were shown to be more accurately predicted using bagging and random forest than boosting models [14,16], but the performance may depend on the study settings including the amount of data, target/feature variables, hyperparameters, and evaluation measures. Whereas the prediction performance of RF has been reported in previous studies, boosting-based XGBoost, in which a sequential procedure focuses on errors between the actual and fitted values from the previous step of the sequence, seemed to be more suitable for predicting land prices in Seoul. However, it was found that the prediction accuracy of XGBoost degraded as the amount of data increased, whereas the prediction accuracy of the RFs improved as the amount of data increased. These findings are likely caused by the fact that the boosting-based models are relatively sensitive to outliers [32]; the 2017–2020 data could include more outliers and variations than the single-year data. In South Korea,

the real estate market has shown a rapid increase in prices since 2019, which might be the reason for the higher accuracy of the single-year data.

Clear patterns were not found in the separate analyses according to land use, but complementary results were obtained by RF and XGBoost models when analyzing the residential areas only. The accuracy of the RF was highest in the residential areas, whereas the accuracy of XGBoost was lowest in that area. Similar to the aforementioned comparison of the results between the 2020 data and the 2017–2020 data, the results of XGBoost models were more accurate on the 2020 data than on the 2017-2020 data when analyzing residential areas only. The real estate market forms different submarkets based on supply and demand drivers that can be determined by temporal periods and specific land uses [38,39]. Housing prices vary both with respect to housing characteristics and location [40], and this might lead to more outliers and variation in the land prices of residential areas. Further analyses also need to consider submarkets based on geographical locality and price variation.

The limitations of this study are as follows. As stated in the review article [41], we also address the limitation of machine learning models in terms of the black box nature and the poor inferential ability. While machine learning algorithms are shown to provide high predictions, it is unclear how to obtain consistent results or determine the best model. The different machine learning algorithms yielded different accuracies and feature importance. The lack of clear guidelines on the application of machine learning reveals limitations in the future applications of models and the resultant analyses. In addition, the different results obtained from the various hyperparameter configurations in machine learning are another limitation. Various methods for tuning hyperparameters exist; however, not all tuning methods were used in this study owing to time constraints. Finally, this study considered various land-use features, but other variables that can influence land price need to be incorporated. As noted earlier, the significance of accessibility to jobs, amenities, or transportation to real estate price has been stated in the literature [24–26]. Future research could incorporate neighborhood characteristics such as social and physical aspects that affect prices, considering real estate localities.

## 5. Conclusions

Research on estimating real estate value has benefitted from the explosion of more accurate machine learning models with publicly available big data. In this study, we automatically refined price data from the massive data collected through public API and used two ensemble machine learning methods, Random Forest and XGBoost, to estimate 52,900 land prices with 24 land-use features in Seoul. The XGBoost results showed overall higher prediction accuracy than that of Random Forest, but in the separate analysis for residential land uses, Random Forest achieved somewhat higher prediction on the 2017–2020 data. Both the XGBoost and Random Forest models identified the most important feature for the land appraisal value, but differed in their importance ranking for the remaining features. This study could not determine the superiority between the two ensemble models, but the limited results require future analysis incorporating various other characteristics of real estate localities, following systematic guidelines on the application of machine learning.

**Author Contributions:** Conceptualization, J.K. and J.W.; methodology, J.K. and J.W.; software, J.K., J.W. and H.K.; validation, J.K., J.W. and J.H.; formal analysis, J.K. and J.W.; investigation, J.K. and J.W.; resources, J.K., J.W. and H.K.; data curation, J.K., J.W. and H.K.; writing—original draft preparation, J.W.; writing—review and editing, J.K., J.W., H.K. and J.H.; visualization, J.W. and J.H.; supervision, J.K. and J.W.; project administration, J.K. and J.W.; funding acquisition, J.K., J.W. and H.K. All authors have read and agreed to the published version of the manuscript.

**Funding:** This work was supported by the National Research Foundation of Korea (NRF) grant funded by the Korean government (2018R1C1B5086305).

**Institutional Review Board Statement:** Not applicable.

**Informed Consent Statement:** Not applicable.

**Data Availability Statement:** Data sharing is not applicable to this article.

**Conflicts of Interest:** The authors declare no conflict of interest.

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
