# Peer review of "Machine-Learning-Based Prediction of Land Prices in Seoul, South Korea"

_sustainability, doi:10.3390/su132313088_

Round 1
Reviewer 1 Report
The authors investigate the use of machine learning for the prediction of land prices in Seoul, South Korea. For this purpose, land use information was obtained from various land- and building-related datasets.
The article is well-structured and meets the specifications of the journal.
The authors have made a thorough presentation of the dataset features summarizing the statistics of the corresponding variables. A similar approach was followed for the presentation of the results.
The Introduction section is focused but is not supported by enough references. Also, at the end of this section, the authors should mention the structure of the article. It should be clearly stated in the Introduction section the novelty of this paper.
The steps of data and variable processing are briefly illustrated in Figure 2. The authors give some useful background information about the employed prediction models. The given details justify the use of machine learning methods for the task of interest. However, the hyperparameters of the RF, XGBoost algorithms would be better to be presented in tabular form. Moreover, the references in this section are not enough to support the relevant arguments. Machine learning models are well-described but not appropriate refereed.
The research outcomes (numerical results) are presented in tables considering some criteria such as the zone type, the year, the features assuming two machine learning models.
The authors should define the accuracy metric and support it with relevant works in machine learning.
The Discussion section captures not only the research findings but also some limitations. However, a comparative analysis should be made with previous research studies in terms of features, models e.t.c. Please highlight the key findings of your work by pointing to similarities and differences of other ongoing studies in this field.
The authors also noted at the last lines of the paper a future direction of their work. They aim to incorporate neighbourhood characteristics in their models. They made a too short reference (lines 372-373), which is insufficient to understand their future purpose. Also, the authors should elaborate more on the possible extensions of this study.
I suggest the authors add a conclusion section to i) report the numerical results of the best performing model, ii) discuss the future works of the current paper and iii) conclude it.
References are not enough and should be enriched considering up-to-date works. To sum up, although the application is interesting, the technical contribution is low as the authors have applied known techniques and methods.
Reviewer 2 Report
This study estimates land prices by combining a variety of land-use data from various land- and building-related datasets. From January 2017 to December 2020, 52,900 land prices in Seoul, South Korea, were estimated using the random forest and XGBoost methods. The models were also trained separately for different land uses and time periods. Overall, the results show that XGBoost helps to improve prediction accuracy. The paper is interesting, while there are some concerns that should be addressed. First of all, the authors provided three research gaps, while I believe some of them can not be considered as a gap, for example, issues that are related to database can not be considered as a gap. Or this contribution “First, most previous studies estimated the transaction or list prices of real estate without separating building and land data.” I believe emphasizing dividing data is not a contribution. Therefore, authors should clearly present the research gaps of the available studies and then state their contributions in the introduction. Furthermore, I think the title of the paper is not appropriate and is a bit unclear. It is better to state the method in the title. In addition, How variables were obtained? There are many different variables in the literature, for example, see:
Nazemi, Behrooz, and Mohsen Rafiean. "Modelling the affecting factors of housing price using GMDH-type artificial neural networks in Isfahan city of Iran." International Journal of Housing Markets and Analysis (2021).
Chen, Yuer, et al. "The impact on neighbourhood residential property valuations of a newly proposed public transport project: The Sydney Northwest Metro case study." Transportation Research Interdisciplinary Perspectives 3 (2019): 100070.
Abidoye, Rotimi Boluwatife, et al. "Impact of light rail line on residential property values–a case of Sydney, Australia." International Journal of Housing Markets and Analysis (2021).
It is better to refer to previous studies to obtain variables. In addition, I think the comparison is not fair. The authors should compare their results with several powerful methods such as MLP, SVM, …. or similar methods which are available in the literature. In addition, section 2.4.2 should be presented in a clearer way. This section is a bit hard for a reader to follow. In addition, many studies that have used artificial intelligence techniques in the literature have been missed in the literature review.
Round 2
Reviewer 1 Report
I think the authors have responded acceptably to the previous comments.
I have no additional remarks on the revised version.
Reviewer 2 Report
Accept.